# *Corynebacterium striatum* Bacteremia during SARS-CoV2 Infection: Case Report, Literature Review, and Clinical Considerations

Andrea Marino [1,2,*], Edoardo Campanella [3], Stefano Stracquadanio [1], Manuela Ceccarelli [2], Aldo Zagami [2], Giuseppe Nunnari [3] and Bruno Cacopardo [2]

1   Department of Biomedical and Biotechnological Sciences, University of Catania, 95123 Catania, Italy; s.stracquadanio@unict.it

2   Department of Clinical and Experimental Medicine, Unit of Infectious Diseases, ARNAS Garibaldi Hospital, University of Catania, 95122 Catani, Italy; manuela.ceccarelli@unict.it (M.C.); aldozagami@alice.it (A.Z.); cacopard@unict.it (B.C.)

3   Department of Clinical and Experimental Medicine, Unit of Infectious Diseases, University of Messina, 98124 Messina, Italy; edo.campanella93@gmail.com (E.C.); giuseppe.nunnari@unime.it (G.N.)

*   Correspondence: andreamarino9103@gmail.com; Tel.: +39-34-6358-9243

**Abstract:** Bacterial infections, especially those in hospital settings, represent a major complication of COVID-19 patients, complicating management and worsening clinical outcomes. *Corynebacterium striatum* is a non-diphtheric actinobacterium that has been reported as being the causative agent of several different infections, affecting both immunocompetent and immunocompromised patients. Recently, *C. striatum* has been recognized as a nosocomial pathogen that is responsible for severe infection in critical patients, as well as in fragile and immunocompromised subjects. *C. striatum* has been described as the etiological agent of bacteremia, central line infections, and endocarditis. We report a case of a 91-year-old woman who was hospitalized due to SARS-CoV-2 infection, who developed *C. striatum* bacteremia and died despite antimicrobial therapy and clinical efforts. Furthermore, we discuss *C. striatum* diagnosis and treatment based on evidence from the scientific literature.

**Keywords:** *Corynebacterium striatum*; Corynebacterial infections; Corynebacterial bacteremia

## 1. Introduction

As the SARS-CoV-2 pandemic persists, bacterial coinfections are reported worldwide as being ascribed to significant morbidity and mortality. Secondary bacterial infections could have occurred in more than 50% of fatal COVID-19 patients [1].

Several studies have analyzed bacterial infections in COVID-19 patients among different countries, reporting a prevalence ranging from 12.4 to 50% [1]. A multicenter analysis from China reported that bacterial coinfections were observed in almost 34.5% of severe/critical COVID-19 patients, regardless of the fact that most of them had received antibiotic treatments [2].

*Corynebacterium (C.) striatum* is a non-diphtheric Gram-positive, facultative anaerobic, catalase negative, asporogenous, non-motile actinobacterium [3]. Despite the fact that it is a skin and mucosal-commensal microorganism [4], in the last few decades, it has been increasingly reported as being the causative agent of a variety of infections in both immuno-compromised patients, such as those in critical conditions, and in immunocompetent hosts. Although several issues subsist regarding its microbiological isolation and antimicrobial testing, as well as the differentiation between infection and contamination, *C. striatum* has been recognized as being a nosocomial emerging pathogen that is associated with several diseases, including cases of bacteremia, endocarditis, and pneumonia [3]. Extended hospitalization, invasive procedures, and the prolonged use of antibiotics represent the major risk factors that correlate with *C. striatum* infections [5].

In this report, we describe the uncommon case of a 91-year-old woman who was hospitalized due to SARS-CoV-2 infection, who developed *C. striatum* bacteremia and died despite antimicrobial therapy and clinical efforts. We also review and analyze the literature evidence on *C. striatum* diagnosis and treatment.

## 2. Case Presentation

A 91-year-old woman was admitted to our ward from a COVID-19 residential care facility because of the onset of fever (T max: 38 °C) associated with urinary symptoms (dysuria). She received a two-dose vaccination against SARS-CoV-2 (the last dose being 8 months before hospital admission; BNT162b2 vaccine; BioNTech Pfizer, Inc., New York, NY, USA).

Her medical history was positive for hypertension, chronic atrial fibrillation, stroke, chronic kidney disease, and thyroidectomy. She had uterine cancer with peritoneal metastasis. The patient took levothyroxine, pantoprazole, furosemide, enoxaparin, and canrenone.

Upon admission, she was in a poor general state with disturbances in consciousness (Glasgow Coma Scale 11), was feverish (T: 37.5 °C), her blood pressure (BP) was 120/60 mmHg, her pulse rate was 70 bpm, and the oxygen saturation in the room air was 98%. Physical examination revealed bilateral leg pitting oedema. A peripheral venous catheter along with a urinary catheter were placed. An electrocardiogram showed alterations compatible with her medical history.

Blood tests showed a normal count for white blood cells ($9.6 \times 10^3/\text{mm}^3$), hemoglobin levels of 11.1 g/dL, and a normal platelet count ($259 \times 10^3/\text{mm}^3$). Her inflammatory markers were elevated: C-reactive protein (CRP) was 7.15 mg/dL (normal range < 0.5 mg/dL), and erythrocyte sedimentation rate (ESR) was 48 mm/h (normal range < 10 mm/h). Procalcitonin was negative. Creatinine was 1 mg/dL (eGFR with CKD-EPI was 49.2 mL/min). Liver markers, including coagulation tests (INR 1), were normal. D-dimer levels were also elevated (1.827 ng/mL). HBV, HIV, and HCV serology tested negative. Lab tests also revealed hypopotassemia (2.7 mEq/L). A urine examination showed leukocyturia, whereas the urine culture was negative (Table 1).

**Table 1.** Laboratory findings at the time of admission, at the time of blood cultures, and at the day of exitus.

| Lab Parameters (Reference Range) | Time of Admission | Time of Blood Cultures | Day of Exitus |
|---|---|---|---|
| WBC, cells/mmc (4000–10,000) | 9600 | 4200 | 1200 |
| Neutrophils, % (40–75) | 82.3 | 87.3 | 84.4 |
| Lymphocytes, % (25–50) | 12.8 | 9 | 11.7 |
| Monocytes, % (2–10) | 4.1 | 2.9 | 1.5 |
| Platelets, cells/mmc $\times 10^3$ (150–400) | 267 | 50 | 31 |
| Haemoglobin, g/dL (12–16) | 11.1 | 8.4 | 7.1 |
| AST, UI/L (15–35) | 35 | 25 | 51 |
| ALT, UI/L (15–35) | 22 | 12 | 10 |
| LDH, UI/L (80–250) | 370 | 324 | 577 |
| Creatinine, mg/dL (0.8–1.2) | 1 | 0.77 | 1.07 |
| $Na^+$, mEq/L (135–145) | 143 | 135 | 129 |
| $K^+$, mEq/L (3.4–5.1) | 2.7 | 2.8 | 4.4 |
| $Cl^-$, mEq/L (98–107) | 100 | 96 | 93 |
| CRP, mg/dL (0–0.5) | 7.15 | 12.29 | 1.25 |
| ESR, mm/h (0–10) | 48 | 80 | 100 |
| Procalcitonin, µg/L (<0.5) | 0.17 | 7.68 | 0.71 |
| INR, (0.8–1.1) | 1 | 1.40 | 1.39 |
| D-dimer, ng/mL (<250) | 1827 | 431 | 1932 |
| Fibrinogen, mg/dL (200–400) | 100 | 100 | 100 |
| Ferritin, ng/mL (20–200) | 220 | >2000 | >2000 |

WBC: White Blood Cells; AST: aspartate aminotransferase; ALT: alanine aminotransferase; LDH: lactic dehydrogenase; CRP: C-reactive protein; ESR: Eritrosedimentation rate; INR: International normalized ratio.

A chest X-ray showed bilateral interstitial thickening without lobar consolidations.

Treatment was commenced with intravenous ceftriaxone 2 g/die, along with IV fluids with potassium supplementation. Ceftriaxone was administered for 6 days, achieving a reduction in inflammatory markers, along with the amelioration of urine analysis parameters. Her fever disappeared within two days of antibiotic administration.

On the 10th day from the time of admission, the patient developed a high fever (T 39 °C) along with hypotension (BP 80/45 mmHg). Her lab tests showed increasing levels of CRP (12.29 mg/dL) and procalcitonin (7.68 µg/L).

Two sets of blood cultures, at the same time and from both the peripheral vein and intravenous catheter, and urine culture were performed, along with substitution therapy with 2 units of fresh frozen plasma and 2 units of red blood cell concentrate, due to a reduction in hemoglobin levels (7 g/dL), as well as decreased platelet counts and fibrinogen levels (PLT: 50,000/mm$^3$; FIB: 100 mg/dL). The INR was 1.40 (Table 1). Empiric antibiotic therapy was switched to intravenous meropenem, 1 g three times daily, plus linezolid 600 mg twice a day. A full body CT scan, performed without contrast due to the patient's impaired renal function, did not show active bleeding.

Within 72 h from the sample collection, peripheral vein blood cultures tested positive (a BD BACTEC culture of aerobic/anaerobic vials and incubation using the BACTEC FX automated blood culture system (Becton, Dickinson and Company, Franklin Lakes, NJ, USA)) for *Corynebacterium striatum*, whereas cultures from the IV catheter produced negative results. An antibiotic susceptibility test was not performed, and empirical antibiotic therapy was continued; whereas, performing the source control, the patient's vascular devices as well as the urinary catheter were substituted and sent for culture, which later produced negative results.

However, the day after the isolation of *C. striatum* from the blood cultures, despite the progressive amelioration of inflammatory markers (CRP levels up to 1.25 mg/dL) and procalcitonin levels (up to 0.71 µg/L), as well as the initiation of noradrenaline infusion, her clinical status continued to deteriorate until exitus occurred. A SARS-CoV2 molecular test repeatedly produced positive results during the patient's hospital stay.

## 3. Discussion

As reported in the literature, the bacteria most commonly isolated in patients who are positive for COVID-19 are *Mycoplasma pneumoniae, Staphylococcus aureus, Streptococcus pneumoniae, Escherichia coli,* and, to a lesser extent, *Klebsiella pneumoniae, Haemophilus influenzae, Pseudomonas aeruginosa, Acinetobacter baumannii,* and *Bordetella pertussis* [1,6,7]. In some cases, bacteremia following respiratory infection has been already described [1]. However, other opportunistic Gram-positive or Gram-negative bacteria, as well as fungi, can be responsible for superinfections in COVID-19 patients. *Corynebacterium striatum* is a nosocomial opportunistic pathogen, and a Gram-positive rod member of the normal human skin microbiota [4]. In the last few decades, *C. striatum* is increasingly being considered to be an emerging pathogen, with most infections being referred to as nosocomial and occurring especially in patients with significant comorbidities or in patients with skin integrity alterations, indwelling devices, and prolonged exposure to broad-spectrum antibiotics [5,8].

Although it was previously postulated that only endogenous infection pathways were possible, in recent years, epidemiological studies have revealed the possibility of corynebacteria patient-to-patient transmission via healthcare-personnel hands [9–11].

*C. striatum* is the causative agent of various illnesses, including skin infections such as cellulitis [12], osteomyelitis [13], meningitis [14], pneumonia [15], bacteremia [9,16–20], and infective endocarditis [21]. Over the past few years, the number of reported cases has been rising due to the increase in the life expectancy of immunosuppressed patients and the improvement in microbiological techniques that are suitable for accurately identifying this microorganism [3].

The isolation of *C. striatum* from culture is often considered as being contamination unless repeated cultures reveal the same result [22]. In a recent systematic review, Milosavl-

jevic et al. stated the reliability of the Vitek 2 system in the identification of *C. striatum* after the inclusion of this pathogen in the database [3], despite the fact that the scientific literature has reported misidentifications when using biochemical methods [23,24]. However, the gold standard technique for *C. striatum* identification is 16S RNA gene sequencing or, alternatively, matrix-assisted laser desorption ionization-time of flight mass spectrometry (MALDI-TOF), due to its simplicity and cost efficiency [3].

Ishiwada et al. [25] reported that only 60% of episodes of *C. striatum* bacteremia are associated with a fever >38 °C, and about half of the infected patients improved without therapy, or even despite inappropriate antibiotic treatment. The organisms in these cases were assumed to be contaminants, whereas only those patients recovering after an appropriate antibiotic course were considered to be truly infected. Yamamuro et al. [26] reported that *C. striatum*, along with *C. jeikeium*, caused true bacteremia more frequently than other Corynebacterium species, defining true *Corynebacterium* spp. as bacteremia cases with at least two sets of positive blood cultures from a patient with signs of infection.

Abe et al. [27] also reported that in patients with hematological disorders, *C. striatum* and *C. jeikeium* were the two major Corynebacterium species causing bloodstream infections.

In addition, Mushtaq et al. [28] showed that the blood culture time for positivity has also been associated with clinical significance, along with the number of positive blood culture sets.

Furthermore, biofilm production is an important virulence factor of *C. striatum*, as has been suggested in several studies [29–31]. Patients with multiple positive blood cultures had significantly higher levels of biofilm formation, postulating that the biofilm phenotype may be associated with the clinical significance of *C. striatum* recovery from patients' samples. Indeed, the association between biofilm formation and *C. striatum* nosocomial outbreaks has been highlighted by De Souza et al. [31]: biofilm would favor bacterial persistence in the hospital environment, enhancing its invasive potential and making the germ more resistant to the action of the immune system, as well as to multiple antibiotics.

This evidence could explain the growing number of bloodstream and catheter-related infections caused by *C. striatum* that have been reported over the last few years. Moreover, considering the role of biofilms in infective endocarditis (IE) [32,33], physicians should consider the possibility of infective endocarditis diagnosis in the case of bacteremia due to non-diphtheric corynebacteria. After all, IE is the most common manifestation of *C. striatum* [21,24,34,35].

Unlike many other infective agents, the automatized systems of identification (Phoenix, Vitek 2) do not perform antimicrobial susceptibility testing (AST) for *C. striatum*, leading the physician to use empirical therapy without the possibility of conducting appropriate antimicrobial stewardship [11]. Moreover, in accordance with the European Committee on Antimicrobial Susceptibility Testing (EUCAST) guidelines, laboratory AST for *C. striatum* requires particular settings as well as the use of horse-defibrinated blood and β-NAD for both broth microdilution, MIC evaluation, and disk diffusion in agar. The incubation time before obtaining the results could be up to 44 h, making the tests cumbersome and time consuming, especially in a hospital environment [36]. For these reasons, *C. striatum* samples are often sent to an external laboratory and, in some cases, the antibiotic susceptibility profile may be communicated to the infectious disease specialist after the patient's discharge [16] or death. Luckily, this species seems to be 100% susceptible to linezolid and vancomycin [3,27,30,37–39]—two antibiotic molecules that cannot be used for prolonged lengths of time, due to their toxicity [27,40,41]—with only one recorded case of *Corynebacterium* spp. being resistant to vancomycin in a clinical case dated back to 1991 [42], and another case of a septic patient, in which the administration of vancomycin did not seem to be effective [16]. Furthermore, although linezolid and vancomycin epidemiological cut-off values (ECOFF) and MIC distribution for wild-type *C. striatum* are not available on the EUCAST site, the correlation between disk diffusion and the MIC values demonstrated that all of the 248 and 258 *Corynebacterium* spp. isolates analyzed for vancomycin and

linezolid susceptibility had an MIC of ≤1 mg/L (vancomycin range ≤0.5–1 mg/L, linezolid range 0.125–1 mg/L) [43]. Notably, the majority of the tested strains were *C. striatum* (78 out of 284 isolates, 27.5%), and the EUCAST breakpoint for both the antibiotics indicate a susceptibility for MIC of ≤2 mg/L [36,37].

Considering the case that we presented, linezolid had been already administered as an empirical therapy, since vancomycin was ruled out due to both its nephrotoxicity and the patient's impaired renal function. Overall, linezolid was administered for 4 days before the exitus, whilst other case reports of *C. striatum* infections have reported on the use of this antibiotic for 1–6 weeks to eradicate the bacterium [20,40].

However, therapeutic failure should be further investigated to reveal whether this is due to the general condition of the patient, the bacterial superinfection, or to the timing of antibiotic administration.

Promising antibiotic alternatives may be represented by tigecycline or daptomycin [44]. Indeed, *C. striatum* seems to be susceptible to daptomycin, whether alone or combined with rifampicin [37]; despite the emergence of high-level daptomycin resistance has been already described [18,45]. In addition, high levels of resistance to a wide range of other antimicrobial drugs, including most β-lactams [11], macrolides [46], fluoroquinolones [47], aminoglycosides [48], and cotrimoxazole [3], have been reported.

The patient's temporary amelioration after treatment with ceftriaxone may suggest a primary bacterial infection, probably from the urinary tract—recovered after treatment with third-generation cephalosporin—not caused by *C. striatum*, this latter likely entered the patient's bloodstream through catheters or indwelling devices after hospitalization.

Overall, although the patient that we presented had no relevant immunosuppression diseases [33,49–51], except for cancer diagnosis, she presumably died due to disseminated intravascular coagulation (CID)-like syndrome caused by septic shock. Despite the SARS-CoV2 infection, the patients did not have any lung involvement; however, we could not definitely rule out viral effects such as hyperinflammation and hypercoagulopathy [52–54]. Therefore, it is hard to define whether the antibiotic therapy had sufficient time to be effective, or whether the multi-organ failure, caused by bacterial and viral co-infection, led to the exitus before or notwithstanding the opportune therapy [55,56].

## 4. Conclusions

This case highlights the need to consider the patient's history of exposure to broad-spectrum antibiotics or chronic diseases to maintain high levels of suspicion toward these kinds of infections. Empirical antibiotic therapy with vancomycin or linezolid should be considered when choosing the appropriate antimicrobial regimen, especially with increasing numbers of reported cases of *C. striatum* bacteremia. Moreover, due to the association between *C. striatum* bacteremia and infective endocarditis, the physician should be aware of the need of consider the diagnosis of IE in order to eventually perform echocardiography. Unfortunately, despite correct management of the infection, we could not successfully treat the patient because of her poor general condition and comorbidity.

In a hospital setting and in emergency cases such as the one reported herein, it is very important to rapidly find the cause of infection and the best therapy to manage it, especially for patients with a high risk of multi-organ failure. Unfortunately, the common presence of *C. striatum* as commensal bacteria leads to the necessity for two positive blood cultures to indicate this bacterium as being the cause of bacteriemia. This could be the first reason for the problematic treatment of these blood infections. Moreover, the correct identification of *C. striatum* could require systems that are not available in every hospital.

**Author Contributions:** Conceptualization, A.M. and E.C.; methodology, M.C.; investigation, A.Z.; data curation, G.N.; writing—review and editing, S.S.; supervision, B.C. All authors have read and agreed to the published version of the manuscript.

**Funding:** This research received no external funding.

**Institutional Review Board Statement:** Not applicable.

**Informed Consent Statement:** Written informed consent has been obtained from the patient to publish this paper.

**Data Availability Statement:** Not applicable.

**Conflicts of Interest:** The authors declare no conflict of interest.

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
