# Peer review of "Corynebacterium striatum Bacteremia during SARS-CoV2 Infection: Case Report, Literature Review, and Clinical Considerations"

_2036-7449, doi:10.3390/idr14030042_

Round 1

Reviewer 1 Report

This is an interesting case report of a 91 year old patient with a history of metastatic uterine cancer transferred from a residential care facility to an acute care hospital for presumed urosepsis (of unknown cause) and expiring due to a hospital-acquired Corynebacterium striatum blood stream infection during the Covid outbreak in Italy. The case raises the important point about the emergence of C. striatum as a nosocomial concern. The following comments are offered in order to strengthen the presentation.

Case report - in general, incomplete.

I am struck by the lack of diagnoses made at key points in the case. Lab values and medications are given and procedures were described, but no diagnoses. What was the admitting diagnosis? Procalcitonin may not always rise in cystitis. Did the patient have cystitis or more systemic infection? You refer to this as a potential primary "super" infection. Either something is lost in translation or this is an inappropriate use of the term superinfection. Did she have two infections on admission?

On day 10, she was septic and hypotensive, but with no ionotropic support given? The patient was overdosed on meropenem (6 g per day) and could have precipitated pancytopenia. Based on her renal function, she should have received 0.5-1 g every 12 hours.

Her haemoglobin decreased by 4 g/dL over 10 days and platelets decreased  by over 230,000. Where was she bleeding?

What was the cause of death?

Literature review and discussion

A reasonable integration of extant literature. More case analysis would be helpful to illustrate the points made throughout. Summation is found at the end, however.

This patient likely died of a nosocomial infection caused by C. striatum complicated by meropenem. With the time delay from the second Covid vaccination, are you suggesting that she also had Covid? Was she tested for Covid? Please explain your thinking to contextualize viral contributions to this case.

Author Response

Reply: Thank you for your precious suggestions and point of view. We believe admission diagnosis was urinary tract infection, due to urine characteristics and initial amelioration with empiric ceftriaxone administration and despite negative urine culture. We added this statement in discussion section, fixing the first “superinfection” term, which was a typo, in “bacterial infection”.

As regards inotropic support, line 99 reports the use of noradrenaline (unfortunately we had very short time to manage patient’s critical conditions, but we tried to do our best). As regards meropenem dose, “2 gr” was a typo/mistake, and we fixed it accordingly in the case section.

As regards the cause of death, we added some extra lines in the paper explaining that probably she died due to disseminated intravascular coagulation (DIC) caused by septic shock. She was not bleeding as confirmed by full body CT scan (performed without contrast); platelet reduction was ascribed to DIC. We added some extra information in the case section.

Finally, considering SARS-CoV2 infection, thorax X-ray did not show respiratory disease as well as patient’s clinical conditions. We repeatedly performed molecular SARS-CoV2, which resulted positive during patient’s hospital stay. We do not believe COVID 19 play a clear role in this case, however we could not definitely rule out viral effects (such as hyperinflammation or hypercoagulability) on the general patient’s conditions. We clarify these points in discussion section.

We are sorry for the missing details, but we focused on the Corynebacterium striatum bacteremia (that was the key-point of our case and the innovation). We hope details we added will confer more clearness to our case.

Reviewer 2 Report

Marino and coauthors present a case study of a woman admitted to hospital for COVID-19 and subsequently moved to an infectious disease ward. This study is of interest to readers and clinicians due to the unusual nature of C. striatum coinfection with SARS-COV-2. It highlights the need to rapidly and accurately identify an infectious agent as well as the antibiotic resistance profile of the agent. In the case of this patient, antibiotic susceptibility testing was not performed, making it difficult to determine whether treatment was ineffective due to resistance or underlying comorbidities complicating treatment.

This manuscript is well-written, although it may be useful to submit it to a native English speaker to make minor edits to sentence structure and syntax simply for ease of reading.

The authors bring up the impact of biofilm formation as a virulence mechanism in C. striatus, and mention that catheter and other devices were submitted for culture. However, they do not report the results of those cultures and whether the organism was also identified there. If these data are available they should be reported in the manuscript, especially considering the discussion of false positive identifications in blood culture.

Finally, it is not clear whether the patient had recovered from SARS-COV-2 infection prior to admission to the infectious disease ward. This should be made explicit, because it would have implications for diagnosis and treatment of a coinfection instead of secondary infection in an immunocompromised patient. The authors hint at the end of the discussion section that it may have been a coinfection, but this should be made explicit if that information is available.

Author Response

Reply: Thank you for your valuable opinion. Patient’s vascular devices and urinary catheter culture resulted negative; we received this information after the patient died. We stated this point within the text.

As regards SARS-CoV2 infection, we added extra details about test results and consideration in both discussion and case section.

Finally, a mother tongue colleague revised the paper as you suggested.

Reviewer 3 Report

Marino et al. present a case study on a COVID-19 patient who developed bacteremia due to Corynebacterium striatum. The authors also discuss the role of C. striatum, as an emerging pathogen in a variety of infections, as well as the obstacles to its detection and the treatment of its disease(s). The manuscript is set well across and properly referenced. I only have some minor comments.

I would suggest that the authors double-check if the bacterial names used in the manuscript are written in italics (ln 2, ln, 26, ln 145, ln 149…). The same is true for References: all the bacterial names are not in italics and the names of the bacterial genes (ln 371, ln 374) as well. Also, there are spelling mistakes (ln 250, ln 262, ln 272…) and reference duplication (compare Refs. 30 and 43; 39 and 48; 20 [also incomplete] and 48).

Ln 18: I would write a non-diphtheric actinobacterium, instead of ‘Gram-positive bacillus’, to be less general about C. striatum’s taxonomic status.

Ln 26: use Corynebacterium/corynebacterial as an adjective for infections or bacteremia

Ln 37-38: ‘Gram-positive’ (not ‘gram-positive’, since Gram is the name of the person who discovered the staining), ‘asporogenous’ (instead of ‘asporigenous’), for ‘bacillus’ see my comment to ln 18.

Ln 41-45: This statement should be referenced.

Ln 48-51: For clarity, the authors can split this paragraph into 2 sentences: the first one ending with ‘… therapy and clinical effort.’, and the second starting with ‘We also review and analyse…’.

Ln 93: leave only one of the double parentheses – ‘))’

Ln 113-114: Spell out Pseudomonas and Acinetobacter since their names are mentioned for the first time.

Ln 120: ‘referred to as’ instead of ‘referred as’

Ln 123-125: This sentence is a little bit unclear. Also, since the authors refer to corynebacteria (or they mean ‘corynebacterial’) in general, ‘Corynebacteria’ should be spelt with a small ‘c’.

Ln 146, 182, 187… : ‘spp.’ should not be in italics since it is not part of the bacterial name

Ln 165-166: change ‘non-diphtheriae corynebacterial’ to ‘non-diphtheric corynebacteria’

Ln 182: Use small letters for the generic names of drugs/antibiotics. Compare Ln 182, 183, 184, 187, 188 to Ln 192, 193.

Ln 229: change ‘a commensal bacteria’ to ‘commensal bacteria’ or ‘a commensal bacterium’

Author Response

Reply: Thank you for your precision and valuable expertise. We changed the text according to all your suggestions.

Round 2

Reviewer 1 Report

The authors have responded to my suggestions. Many thanks and well done!

Author Response

Thank you for your valuable time and consideration.